# Janus Kinase Inhibitors and Coronavirus Disease (COVID)-19: Rationale, Clinical Evidence and Safety Issues

**DOI:** 10.3390/ph14080738

**Published:** 2021-07-28

**Authors:** Milo Gatti, Eleonora Turrini, Emanuel Raschi, Piero Sestili, Carmela Fimognari

**Affiliations:** 1Pharmacology Unit, Department of Medical and Surgical Sciences, Alma Mater Studiorum—Università di Bologna, Via Irnerio 48, 40126 Bologna, Italy; milo.gatti2@unibo.it; 2SSD Clinical Pharmacology, IRCCS Azienda Ospedaliero Universitaria Sant’Orsola, 40126 Bologna, Italy; 3Department for Life Quality Studies, Alma Mater Studiorum—Università di Bologna, C.so D’Augusto 237, 47921 Rimini, Italy; eleonora.turrini@unibo.it; 4Department of Biomolecular Sciences (DISB), Università degli Studi di Urbino Carlo Bo, Via I Maggetti 26, 61029 Urbino, Italy; piero.sestili@uniurb.it

**Keywords:** coronavirus disease (COVID)-19, janus kinase (JAK) inhibitors, baricitinib, ruxolitinib, real word evidence, adverse effects, drug interaction

## Abstract

We are witnessing a paradigm shift in drug development and clinical practice to fight the novel coronavirus disease (COVID-19), and a number of clinical trials have been or are being testing various pharmacological approaches to counteract viral load and its complications such as cytokine storm. However, data on the effectiveness of antiviral and immune therapies are still inconclusive and inconsistent. As compared to other candidate drugs to treat COVID-19, Janus Kinase (JAK) inhibitors, including baricitinib and ruxolitinib, possess key pharmacological features for a potentially successful repurposing: convenient oral administration, favorable pharmacokinetic profile, multifunctional pharmacodynamics by exerting dual anti-inflammatory and anti-viral effects. Baricitinib, originally approved for rheumatoid arthritis, received Emergency Use Authorization in November 2020 by the Food and Drug Administration in combination with remdesivir for the treatment of COVID-19 in hospitalized patients ≥ 2 years old who require supplemental oxygen, invasive mechanical ventilation, or extracorporeal membrane oxygenation. By July 2021, the European Medicines Agency is also expected to issue the opinion on whether or not to extend its use in hospitalised patients from 10 years of age who require supplemental oxygen. Ruxolitinib, approved for myelofibrosis, was prescribed in patients with COVID-19 within an open-label Emergency Expanded Access Plan. This review will address key milestones in the discovery and use of JAK inhibitors in COVID-19, from artificial intelligence to current clinical evidence, including real world experience, and critically appraise emerging safety issues, namely infections, thrombosis, and liver injury. An outlook to ongoing studies (ClinicalTrials.gov) and unpublished pharmacovigilance data is also offered.

## 1. Introduction

The emergence and rapid spread of the novel coronavirus disease (COVID-19) pandemic is posing a serious challenge to global public health. Several efforts have been invested to prevent the development of the infection, counteract the progression of the disease, and reduce its mortality rate. Although much has been learned, at present, only a few therapeutic strategies have demonstrated actual benefit on hard clinical endpoints such as mortality, mechanical ventilation, length of hospital stay, or time to resolution of symptoms, for instance the antiviral remdesivir and dexamethasone [1].

Drug repurposing, also known as repositioning or rediscovery, allows the identification of novel candidate drugs already approved by the regulatory agencies with a positive benefit/risk assessment for other indications [2]. This strategy facilitates the process of drug discovery and development, saving cost and time without the need for pre-clinical studies [3]. A common molecular pathway represents the basis of drug repositioning [2], and this strategy is often considered a serendipitous approach, as exemplified by the case of aspirin, originally conceived as an analgesic and subsequently repurposed as an antiplatelet drug [4], or sildenafil, initially indicated for the treatment of hypertension and angina and actually used for the treatment of erectile dysfunction [5]. Therefore, drug repurposing represents a promising strategy to satisfy the urgent need to accelerate the identification of COVID-19 effective treatments.

Currently, drugs considered for repurposing for COVID-19 can be categorized into: (i) drugs potentially inhibiting the lifecycle of the virus or (ii) drugs potentially counteracting the effects of severe acute respiratory syndrome coronavirus 2 (SARS-CoV-2) infection [2]. Although drug repositioning certainly shortened the development of new medications to face the pandemic, it nonetheless requires funding, time, and the safety assessment for the new therapeutic indication [6]. Moreover, this strategy may include the optimization of the pharmaceutical formulation and the route of administration, which can significantly improve the risk/benefit profile of the repurposed drug. Indeed, pulmonary administration may favor the accumulation of the repurposed drug in the lung and improve safety by minimizing systemic side effects, in a complementary approach with the vast majority of COVID-19 drugs that are administered systemically [6]. Antiviral, antibacterial, anti-inflammatory, and immune modulator drugs are under investigation in several clinical trials enrolling patients with SARS-CoV-2 infection, although, at present, data on their effectiveness are still inconclusive and inconsistent for effective evidence-based recommendations [7].

Janus kinase (JAK) inhibitors represent promising medicines targeting JAKs to treat inflammatory and immune diseases [8]. As compared to other candidate drugs to treat COVID-19, JAK inhibitors possess key pharmacological features for a potentially successful repurposing: convenient oral administration, favorable pharmacokinetic profile, and multifunctional pharmacodynamics by exerting dual anti-inflammatory and anti-viral effects [9].

In this review, we gained insight into the role of JAK inhibitors for treatment of COVID-19 disease, offering an outlook on the most recent clinical evidence. Additionally, we provided a critical appraisal on the emerging safety issues, namely infections, thrombosis, and liver injury, and the clinical relevance of drug–drug interactions between JAK inhibitors and other agents used to contrast COVID-19 disease.

To this purpose, a literature search on Pubmed/Medline was conducted as of 4 June 2021, combining different keywords (“baricitinib”, “ruxolitinib”, “Janus kinase inhibitors”, “COVID-19”, “severe acute respiratory syndrome coronavirus”, “adverse effects”, “thrombosis”, “liver injury”, and “infection”). Additional data were obtained from Regulatory documents, especially the summary of the product characteristics (SPC), from the European Medicines Agency (EMA), the US Food and Drug Administration (FDA), and the Italian Regulatory Agency (AIFA). The website ClinicalTrials.gov was also queried to highlight unpublished evidence.

## 2. Pharmacological Aspects of JAK Inhibitors

JAK inhibitors are recent drugs indicated for the treatment of inflammatory diseases, such as moderate to severe rheumatoid arthritis, psoriatic arthritis, ulcerative colitis, and myeloproliferative malignancies, including myelofibrosis and polycythemia vera [8]. These diseases are characterized by the aberrant activation of the JAK-STAT (signal transducer and activator of transcription) signaling pathway that first plays a pivotal role in the orchestration of the immune system response. In addition, JAK/STAT signaling cascade is involved in the control of cell proliferation and survival [9]. Different cytokines are involved in the activation of JAK/STAT, including pro-inflammatory cytokines such as interleukin (IL)6, IL12, IL23, and tumor necrosis factor (TNF)α; the anti-inflammatory cytokines IL4 and IL10; hematopoietic cell growth factor, such as the granulocyte stimulating factor; and the metabolic cytokines leptin and ghrelin [10]. In detail, the JAK/STAT signaling cascade is initiated by the cytokine binding to its receptor, which triggers modification in the receptor conformation, leading to the activation of JAKs (JAK1, JAK2, JAK3, and tyrosine kinase 2 (TYK2)). JAKs are a subgroup of non-receptor tyrosine kinases that phosphorylate themselves, becoming enzymatically active and creating a docking site for STAT proteins (STAT1, STAT2, STAT3, STAT4, STAT5a and b, and STAT6). STATs are also phosphorylated, triggering the activation of STAT multimers that translocate into the nucleus and, through epigenetic regulation, activate gene transcription [10]. The different recombination of JAK and STAT subtypes determine the specificity of the final transcriptional effect.

Thanks to their ability to act in the orchestration of the key JAK/STAT intracellular signaling, and not least their reduced manufacturing cost compared to biologic drugs, JAK inhibitors represent an emerging treatment. From the therapeutic point of view, the different selectivity of JAK inhibitors for some JAKs results in different therapeutic indications. Indeed, dysfunction of (i) JAK1 may be implicated in lymphoid neoplasm, (ii) JAK2 in myeloproliferative neoplasm, (iii) JAK3 in severe combined immunodeficiency, and (iv) STAT3 may be associated with hyper-IgE syndrome when deficient, or obesity when chronically activated [10]. Several JAK inhibitors are approved by the FDA or EMA (Table 1) and others are currently under evaluation, such as peficitinib and pacritinib.

All JAK inhibitors share the advantage of orally administration and a favorable pharmacokinetic profile: short half-lives, except for ruxolitinib (114 h terminal t_1/2_); low plasma protein binding; and minimal interference with CYP450-mediated biotransformation pathway. The parental drugs are mainly responsible for the pharmacologic activity (metabolites are approximately 10-fold less active). JAK inhibitors are mostly eliminated renally as intact drugs with the occurrence of partial or minor hepatic metabolism [11]. Slight pharmacokinetic differences were observed among JAK inhibitors, and these aspects will be extensively covered when discussing drug interactions and safety profile. In summary, the disposition of baricitinib was altered in combination with certain transporter inhibitors (i.e., probenecid, organic anion transporter (OAT 3 inhibitor)) compared to tofacitinib or upadacitinib. Moreover, baricitinib and tofacitinib dosages need adjustment in case of renal impairment, unlike upadacitinib, and only tofacitinb requires dose adjustment in patients with hepatic impairment [11].

## 3. JAK Inhibitors and SARS-CoV-2: Rationale and Regulatory Affairs

Apart from the aforementioned serendipitous discovery, computational strategies significantly contribute to drug repurposing. Artificial intelligence (AI) technologies were used to identify old drugs with potential activity to counteract Coronavirus replication and infection. The pandemic represents a challenging opportunity for introducing advanced AI algorithms combined with network medicine for drug repurposing [12]. Benevolent AI’s knowledge graph is a large repository of structured information, including connections extracted from scientific literature by machine learning [12].

Through this approach, AI-derived knowledge graph identifies baricitinib as inhibitors of clathrin-mediated endocytosis of the virus, thereby SARS-CoV-2 viral infection of AT2 lung cells (Figure 1). The AT2 alveolar cells are particularly prone to viral infection, and SARS-CoV-2 uses the ACE2 receptor to infect the cells via receptor-mediated endocytosis [13]. The targets of JAK inhibitors are AAK1 (AP2-associated protein kinase 1) and GAK (cyclin G-associated kinase), pivotal regulators of endocytosis, both members of the numb-associated kinase (NAK) family [14]. Interestingly, among JAK inhibitors, baricitinib showed particularly high affinity for AAK1, and is the only one able to efficiently inhibit AAK1 and GAK at therapeutic concentrations. Ruxolitinib and fedratinib exceed the currently tolerated dose to reduce the viral infectivity, whereas tofacitinib showed no detectable inhibition of AAK1 [15].

By means of cytokines release in response to the SARS-CoV-2, JAK inhibitors are also suggested as a promising approach in patients with COVID-19 for their anti-inflammatory and immunomodulatory activity (Figure 1). Indeed, the inhibition of JAK enzymes, which mediate signaling of pro-inflammatory cytokines including IL6, may mitigate the effects of cytokines released in response to the virus, and limit the damage in patients with severe disease [16]. Among JAK inhibitors, baricitinib 4 mg/day exerted a dual action characterized by the inhibition both of viral endocytosis into the cells and the cytokine outbreak, showing promising effects for the favorable clinical outcome of COVID-19 patients [17]. In detail, baricitinib was designed to selectively inhibit JAK1 and JAK2, with less potency for JAK3. The sparing of JAK3 was suggested to contribute to the reduction in immunosuppression associated with pan-JAK inhibition. However, baricitinib’s purported selectivity is only evident in cell-free assays but not recapitulated in cell-based assays. Baricitinib 50% inhibitory concentrations (IC_50_) for JAK complexes that mediate signaling for several cytokines implicated in COVID-19 immunopathology generally fall below the free maximum plasma drug concentration (Cmax) values achieved with approved dosing [18]. Moreover, baricitinib’s inhibitory effect on IFN signaling could be a double-edge sword. On one hand, it may confer an additional antiviral mechanism. On the other hand, blocking IFN response may be detrimental for host antiviral defenses. In fact, a recent work revealed that type I IFN, and to a lesser extent type II IFN, upregulates ACE2 expression in multiple human cell lines, including upper airway epithelial cells and primary bronchial cells. Therefore, suppressing type I IFN antiviral response could be beneficial, in theory, by interfering with the SARS-CoV-2 replication [19]. However, ACE2 has a protective effect against organ damage related to the renin-angiotensin-aldosterone system, including acute lung injury [20]. Therefore, the net effect of IFN suppression (beneficial versus detrimental) in the setting of COVID-19 might depend on the underlying immune status of the patient and the stage of infection/disease.

In the COVID-19 pandemic, clinical evidence plays a key role not only for the necessary evaluation of the risk/benefit balance of baricitinib, or in general of repurposed drugs, but also for the regulatory perspective, which has to consider the speed of drug approval and the urgence of clinical needs in the rapid evolving of the pandemic [2]. Repurposed drugs could receive provisional marketing approval based on limited clinical data (e.g., after completing only a single clinical trial) with a substantiated benefit/risk balance assessment. This was the case of remdesivir, which received Exceptional Approval by Japan, Emergency Use Authorization by FDA, Conditional Marketing Authorization by EMA or Early Access to Medicine Scheme by the Medicines and Healthcare products Regulatory Agency [2]. In particular, EMA authorized its conditional approval to treat COVID-19 in adults and adolescents with pneumonia requiring supplemental oxygen under the new accelerated pathway [2,6].

In this regulatory scenario, baricitinib received Emergency Use Authorization in November 2020 by the FDA, in combination with remdesivir, for the treatment of COVID-19 in hospitalized patients ≥ 2 years old who require supplemental oxygen, invasive mechanical ventilation, or extracorporeal membrane oxygenation. By July 2021, EMA is also expected to issue the opinion on whether or not to extend its use in hospitalized patients from 10 years of age who require supplemental oxygen. Ruxolitinib was prescribed in patients with COVID-19 within an open-label Emergency Expanded Access Plan. Moreover, its use was authorized by AIFA within a compassionate program since April 2020 (https://www.aifa.gov.it/en/programmi-di-uso-compassionevole-COVID-19, accessed on 4 June 2021).

Although speeding up the process of authorization represents a valuable instrument during a pandemic, solid clinical evidence, including real world data, is necessary to monitor the post-licensing safety and effectiveness of the repurposed drugs to grant a full Marketing Authorization. 

## 4. Clinical Evidence with JAK Inhibitors in COVID-19 Settings

### 4.1. Baricitinib

Overall, 11 studies (one randomized controlled trial, eight observational studies, one case series, and one case report) investigating the efficacy and safety of baricitinib in moderate-severe COVID-19 pneumonia were retrieved [14,17,21,22,23,24,25,26,27,28,29]. The main features of the included studies are shown in Table 2. A total of 885 patients receiving baricitinib were included.

Kalil et al. [28] in the ACTT-2 trial randomized 1033 hospitalized patients affected by COVID-19 pneumonia (515 treated with baricitinib 4 mg/day for two weeks associated to remdesivir versus 518 receiving remdesivir alone). Patients treated with the combination therapy showed a shorter median time to recovery (7 days versus 8 days; relative risk (RR) 1.16; 95% confidence interval ((CI) 1.01–1.32; *p* = 0.03). Median time to recovery was also significantly shorter in the subgroup of patients receiving non-invasive ventilation or high-flow oxygen and treated with baricitinib plus remdesivir (10 days versus 18 days; RR 1.51; 95%CI 1.10–2.08). Although no significant difference in 14-day and 28-day mortality rate was found between the two groups (the study was not powered to detect such a difference), the incidence both of new use of oxygen (−17.4%; 95%CI −31.6 to −2.1) and invasive ventilation or extracorporeal membrane oxygenation (ECMO) (−5.2%; 95%CI −9.5 to −0.9) was significantly lower in patients receiving baricitinib compared to placebo. Furthermore, serious adverse events (AEs) and occurrence of new infections was significantly lower with baricitinib compared to placebo. This lower incidence of nosocomial infections may be related to different mechanisms, including the ability to reduce inflammatory-mediated lung injury and increase lymphocyte counts, its antiviral properties, or its associated shorter recovery time and faster clinical improvement.

Stebbing et al. [25] prospectively assessed 83 patients affected by COVID-19 pneumonia treated with baricitinib 2–4 mg/day, matching them in a propensity score analysis with 83 patients receiving standard of care (hydroxychloroquine, lopinavir/ritonavir, and corticosteroids). Primary composite outcome included occurrence of death or mechanical ventilation. Significantly lower mortality rate or requirement for mechanical ventilation was found in patients treated with baricitinib compared to standard of care (16.9% versus 34.9%; *p* < 0.001). Notably, baricitinib was independently associated as a protective variable with the primary outcome at multivariate regression analysis (hazard ratio (HR) 0.29; CI 0.15–0.58; *p* < 0.001). Hasan et al. [27] reported significant lower intensive care unit (ICU) admission, median length of hospital stay, and median days required to stop the need of supplement oxygen in 20 patients receiving baricitinib 4 mg/day after a loading dose (LD) of 8 mg compared to 17 subjects in which LD was not administered. However, no difference in 30-day mortality rate was found between the two groups. Interestingly, Cantini et al. [22] reported a lower 14-day mortality rate (0.0% versus 6.4%; *p* = 0.01) and lower ICU admission (0.88% versus 17.9%; *p* = 0.019) in 113 patients affected by moderate COVID-19 pneumonia and treated with baricitinib 4 mg/day for 14 days in association with lopinavir/ritonavir, compared to 78 subjects receiving combination therapy including hydroxychloroquine and lopinavir/ritonavir. Similarly, Bronte et al. [21] found a lower mortality rate (5% versus 45%; *p* < 0.001) in 20 patients treated with baricitinib (8 mg/day LD for two days followed by 4 mg/day for additional 7 days) in association with standard of care compared to 56 patients receiving standard of care alone (including hydroxychloroquine and lopinavir/ritonavir). However, both studies were retrospective and no multivariate analysis or adjustment for confounders were performed.

On ClinicalTrials.gov, 20 trials investigating baricitinib in COVID-19 were found, of which two have been completed. However, no additional findings could be extracted. Interestingly, a multi-arm multi-stage randomized adaptive trial (AAMMURAVID trial), assessing the efficacy of baricitinib and remdesivir in association with dexamethasone, has been recently started and supported by AIFA (ClinicalTrials.gov Identifier: NCT04832880).

In summary, baricitinib improved time to recovery in combination with remdesivir and, intriguingly, the combination was associated with fewer serious AEs such as infections and thrombosis, as compared to remdesivir alone. However, the substantial proportion of patients with transaminase increase in more than one trials cannot be disregarded in its safety evaluation.

### 4.2. Ruxolitinib

Overall, 14 studies (one randomized controlled trial, five observational studies, and eight case reports) investigating the efficacy and safety of ruxolitinib in moderate-severe COVID-19 pneumonia were retrieved [30,31,32,33,34,35,36,37,38,39,40,41,42,43]. Main features of included studies are shown in Table 3. A total of 135 patients receiving ruxolitinib were included.

Cao et al. [31] randomized 43 hospitalized patients affected by severe COVID-19 pneumonia (22 treated with ruxolitinib 5 mg × 2/day associated with standard of care versus 21 patients receiving standard of care alone). No significant difference in 28-day mortality rate (0.0% vs. 14.3%; *p* = 0.23) and median time to clinical improvement (12 versus 15 days; *p* = 0.15) was reported between the two groups. Patients receiving ruxolitinib showed a significant radiological improvement at 14-day (90.0% versus 61.9%; *p* = 0.0495). No difference in serious AEs was also noted. In an observational prospective study, Giudice et al. [36] found a significant improvement in median partial pressure of oxygen (PaO2) (*p* = 0.026) and PaO2/FiO2 (fraction of inspired oxygen) (*p* = 0.0395) in patients treated with ruxolitinib in association with eculizumab compared to the best available therapy. However, no significant difference in mortality rate and duration of hospitalization was reported. Several observational studies [32,41,42] reported a rapid clinical improvement in 80–90% of patients with moderate-severe COVID-19 pneumonia, although no comparison with other repurposed agents was performed. Interestingly, Gaspari et al. [35] reported two cases of severe skin AEs with ruxolitinib in a COVID-19 setting.

On ClinicalTrials.gov, 22 trials investigating ruxolitinib in COVID-19 were found, of which three have been completed. Notably, in the RUXCOVID trial (ClinicalTrials.gov Identifier: NCT04362137), 432 patients with COVID-19 associated cytokine storm were randomized: 287 patients treated with ruxolitinib 5 mg twice daily for 14 days with possible extension of treatment to 28 days were compared to 145 patients receiving placebo for 14 days with possible extension of treatment to 28 days. No difference in primary outcome (a composite outcome including death, development of respiratory failure requiring mechanical ventilation, or ICU admission) was found at 28 days.

Unlike baricitinib, mainly case reports are available for ruxolitinib, thus requiring more solid evidence from ongoing clinical trials to definitely assess its benefit/risk profile.

## 5. Safety issues

### 5.1. Safety Profile of JAK Inhibitors in Immune-Mediated Diseases

In the COVID-19 era, pharmacovigilance plays a crucial role for real-time safety monitoring of pharmaceuticals: the global vaccination campaign exemplifies the importance of post-marketing studies for timely detection of rare but serious AEs such as myocarditis, thrombocytopenia, and cerebral venous sinus thrombosis, which may not be fully appreciated in the pre-marketing setting, especially for medications receiving accelerated conditional approval through the so-called rolling review [44]. In particular, spontaneous reporting databases, such as the FDA Adverse Event Reporting System (FAERS) and WHO-Vigibase, have been successfully exploited to characterize the safety profile of drugs, thus informing clinical practice for proactive monitoring [45,46].

The safety of JAK inhibitors is a topic of current interest, with remarkable implications in the treatment of COVID-19. Despite differences in selectivity, a large overlap exists in their safety profiles. Similarly to biological therapy, JAK inhibition can lead to serious and opportunistic infections, and viral infections seem to be particularly frequent [47]. The latest systematic review by Olivera et al. [48] gathered AEs from interventional and observational studies of JAK inhibitors (tofacitinib, filgotinib, upadacitinib, and baricitinib) across major immune-mediated diseases (inflammatory bowel disease, rheumatoid arthritis, psoriasis, and ankylosing spondylitis). Data from 82 studies regarding occurrence of AEs, serious AEs, and AEs of special interest (i.e., infections, serious infections, herpes zoster, malignancy, and cardiovascular events) were synthesized in the review: only an increased risk of herpes zoster was found. These findings were recently confirmed by a post-marketing pharmacovigilance study in FAERS, which found a considerable reporting of infections with baricitinib, including unexpected opportunistic infections such as *Pneumocystis jirovecii* pneumonia [49]. Increased reporting of typical and atypical mycobacterial infections emerged in a similar FAERS analysis on ruxolitinib, thus suggesting the importance of screening and monitoring patients for latent infections prior and during treatment with JAK inhibitors [50]. Although no malignancy signals have been identified to date, long-term follow-up and further research are needed to assess the risk of malignancy associated with these compounds.

### 5.2. Cardiovascular Risk with JAK Inhibitors in Immune-Mediated Diseases

Thromboembolic risk with JAK inhibitors was largely investigated (and debated) since clinical development. In April 2017, the FDA expressed concern about imbalance in thromboembolic events (deep venous thrombosis and pulmonary embolism) observed in placebo-controlled clinical trials of baricitinib [51]. In particular, a growing body of evidence indicated that JAK inhibitors adversely affect several cardiovascular risk factors such as serum lipid profile and platelet count, thus potentially increasing thrombotic risk. This prompted the FDA to finally restrict approval of baricitinib only to the lower 2 mg dose. An updated systematic review with meta-analysis of randomized controlled trials did not reveal a significant increased risk of major adverse cardiovascular events (MACE) and venous thromboembolism in patients with rheumatoid arthritis initiating a JAK inhibitor treatment, as compared to placebo, at least in the short term, although a dose-dependent increase in MACE and venous thromboembolism was observed for baricitinib, thus supporting relevant regulatory measures [52]. In July 2019, the FDA also posted new warnings about an increased risk of blood clots and death with the 10 mg twice daily dose of tofacitinib, which is approved only in patients with ulcerative colitis. Although a systematic review concluded that treatment with biologic or targeted synthetic disease-modifying anti-rheumatic drugs did not significantly increase MACE [53], the regulatory Agencies imposed the sponsor to conduct a dedicated post-authorization safety study. Notably, in early 2021, both the FDA (https://www.fda.gov/drugs/drug-safety-and-availability/initial-safety-trial-results-find-increased-risk-serious-heart-related-problems-and-cancer-arthritis, accessed on 4 June 2021) and EMA (https://www.ema.europa.eu/en/medicines/dhpc/xeljanz-tofacitinib-initial-clinical-trial-results-increased-risk-major-adverse-cardiovascular, accessed on 4 June 2021) warned about initial trial results showing an increased risk of MACE and malignancies (excluding nonmelanoma skin cancer) with use of tofacitinib in rheumatoid arthritis relative to TNF-alpha inhibitors.

Notably, there are a number of complexities involved in the risk assessment of thromboembolism with JAK inhibitors, including the baseline increased thrombotic risk by the underlying disease such as rheumatoid arthritis, and the fact that clinical features of thromboembolism, which range from venous thrombosis to pulmonary embolism, are variable and overlap with some of the clinical features of rheumatoid arthritis [54]. Moreover, the underlying mechanism is still unclear due to a potential “Janus effect” of JAK inhibition that is responsible for a reduction in cytokine-driven thromboembolic events, but also for an increase in prothrombotic potential, as indicated by real-world evidence. There is uncertainty on whether or not selectivity versus JAK1 or JAK2 plays a critical role in risk modulation [55].

These uncertainties surrounding the cardiovascular safety of JAK inhibitors further strengthen the importance of post-marketing data to monitor, detect early, and better characterize the cardiovascular spectrum and clinical features of JAK inhibitors. Accordingly, several pharmacovigilance studies, using FAERS and Vigibase, have recently described the thromboembolic reporting with JAK inhibitors in the real world [56,57,58]. Taken together, these findings provide strong evidence that thromboembolism may be considered a class effect of JAK inhibitors. Therefore, JAK inhibitors should be used with caution in patients with known risk factors for venous thromboembolism. Moreover, taking into account the results of the aforementioned initial trials requested by regulatory Agencies, the recommendation of avoiding JAK inhibitors in high-risk patients does appear to have a rationale.

### 5.3. Safety Issues with JAK Inhibitors of Interest in the Setting of COVID-19

In this section, we specifically discuss the data and pharmacological aspects dealing with the safety of JAK inhibitors in COVID-19, including the aforementioned risks of infections, liver injury, and cardiovascular events such as myocarditis, pro-arrhythmia, and thrombosis, which can also arise from potential drug interactions.

As regards liver injury, increase in transaminases levels was one of the most frequently observed AEs in clinical trials on JAK inhibitors. These data should not be overlooked considering that (i) baseline serum transaminase levels five times above the upper limit were used as exclusion criterion, and (ii) a large prevalence of liver injury was observed in COVID-19 and associated with increased mortality [59]. From a pharmacological standpoint, JAK inhibitors are likely to possess structural alerts and generate reactive metabolites, which have a recognized role in the occurrence of idiosyncratic liver injury [60]. Of note, baricitinib does not possess physiochemical and pharmacokinetic features known to be implicated in liver injury; the drug is not highly lipophilic, only marginally metabolized by CYP3A4, and it did not inhibit OATP1B1, glycoprotein P (P-gp), and breast cancer resistance protein (BCRP), nor bile salt export pump (BSEP), a key transporter involved in cholestasis. These data call for increased awareness by clinicians that idiosyncratic hepatotoxicity (possibly cholestatic) may occur with JAK inhibitors, especially in susceptible patients with COVID-19 (e.g., females with preexisting liver disease/impairment and receiving hepatotoxic drugs).

The likelihood of clinically relevant drug–drug interactions (DDIs) represents an additional safety issue, considering that novel and/or repurposed agents used in the management of COVID-19 are commonly used in combination therapy. These DDIs may potentially result in increased toxicity or underexposure, leading to lack of efficacy. Furthermore, age, underlying comorbidities (e.g., renal or hepatic impairment), other concomitant medications, and acute pathophysiological conditions (e.g., cytokine storm commonly found in late COVID-19 stage) should be considered in the assessment of DDIs between JAK-inhibitors and other COVID-19 agents [61].

Both pharmacokinetic and pharmacodynamic DDIs may occur. Baricitinib and ruxolitinib exhibit subtle but important differences in terms of pharmacokinetic properties [11,62]. The potential of relevant DDIs for baricitinib is low, considering its limited hepatic metabolism (only less than 10% of the drug undergoes biotransformation via CYP3A4), and the lack of inhibitory or inducer activity on CYP450 [11]. Furthermore, baricitinib is a substrate both of BCRP and OAT3, while exhibiting a weak inhibitory activity on BCRP and OAT1/3 transporters only in vitro [11,62]. Conversely, ruxolitinib exhibits extensive hepatic metabolism via CYP3A4 and 2C9, thus being involved in relevant DDIs with CYP450 perpetrators [63]. Additionally, ruxolitinib in vivo exhibits a weak inhibitory activity on P-gp and BCRP [62]. Risk occurrence of clinically relevant DDIs between JAK-inhibitors and other COVID-19 agents (namely remdesivir, dexamethasone, colchicine, tocilizumab, and favipiravir) are reported in Table 4, according to a dedicated online tool (https://www.covid19-druginteractions.org/, accessed on 4 June 2021). The co-administration, both of baricitinib and ruxolitinib, with IL6 inhibitors (e.g., tocilizumab, siltuximab) should be avoided in order to prevent the risk of additive immunosuppression and consequent occurrence of severe invasive bacterial or fungal infections. An increased risk for colchicine overexposure and consequent life-threatening adverse events may occur with the concomitant use of ruxolitinib, particularly in patients affected by renal or hepatic impairment, due to the inhibitory activity of ruxolitinib on colchicine metabolism through P-gp pathway. Favipiravir could increase baricitinib exposure through its moderate inhibitory activity on OAT3, although the clinical relevance of this observed DDI is unknown. Finally, no clinically relevant DDIs are expected with the concomitant use of JAK inhibitors and remdesivir or dexamethasone. 

As regards pharmacodynamic DDIs, it is crucial to place the thromboembolic risk in the context of other cardiovascular issues with drugs used in COVID-19 (Table 5). Of note, the combination with remdesivir is particularly challenging, and the spectrum of cardiovascular events may include hypotension with bradycardia (likely related to remdesivir) and thromboembolism.

### 5.4. Adverse Events of Clinical Interest with Baricitinib from Pharmacovigilance

In order to provide a general picture of the current safety profile of JAK inhibitors in COVID-19, we used the COVID-19 EUA section of the FAERS public dashboard to extract AEs of clinical interest (https://www.fda.gov/drugs/questions-and-answers-fdas-adverse-event-reporting-system-faers/fda-adverse-event-reporting-system-faers-public-dashboard, accessed on 19 May 2021): 91 de-duplicated AEs were reported with the use of baricitinib for COVID-19 therapeutic indications. The analysis was performed according to the traditional pharmacovigilance practices, with a focus on AEs of clinical interest (i.e., rare but serious AEs with high drug-attributable components such as cardiotoxicity and hepatotoxicity) [64]. It is important to remind readers of the inherent limitations of this analysis, including data quality (e.g., missing information and lack of full clinical data), the likelihood of under-reporting, the potential influence of external factors (e.g., media attention), and the lack of exposure data (drug prescription/consumption), which do not allow us to infer causality, incidence, and toxicological risk.

Mean age was 66.1 ± 13.2 years, while no gender preponderance was found (52.3% male); 93.4% of the AEs were serious, namely resulting in death, life-threatening conditions, hospitalization (initial or prolonged), required intervention, or disability. A case-by-case analysis is shown in Table 6. Bacterial or fungal infections were reported in 25 patients, followed by nephrotoxicity (16 cases) and thromboembolic events (13 patients). Hepatotoxicity and cardiotoxicity were, respectively retrieved in nine and eight patients. Mean age ranged from 59.4 years for cases concerning infections to 65.2 years for thromboembolic events, with no gender preponderance. All reports of AEs of clinical interest were classified as serious. Proportion of death ranged from 0.0% for hepatotoxicity to 75% for cardiotoxicity. Only one case of drug-induced liver injury was reported, while no case of QT prolongation or torsade des pointes was retrieved. Remdesivir and dexamethasone were concomitantly reported with baricitinib for COVID-19 management in most of the cases. It is important to underline that several AEs (e.g., transaminases increase, acute kidney injury, thromboembolic events, septic shock) could be directly attributable to complications of severe forms of COVID-19 (disease-related AEs) rather than to baricitinib per se, also considering concomitant medications (e.g., hypotension and bradycardia are likely ascribable to remdesivir), and patient-related risk factors (of note, obesity appears to be a common feature in our data).

## 6. Conclusions and Perspectives

The search for effective treatment of COVID-19 is still an unmet clinical need, and only a limited number of medications have demonstrated a potential benefit on hard clinical endpoints. JAK inhibitors possess key pharmacological features for a potentially successful repurposing, especially considering their dual anti-inflammatory and anti-viral effects. The evidence gathered so far, especially from the ACTT-2 trial, supports the use of baricitinib for up to 14 days, in association with remdesivir, in the gray zone of patients with pneumonia receiving oxygen support without invasive mechanical ventilation [65].

Intriguingly, on 16 June 2021 (i.e., after our literature search), a randomized, double-blind, placebo-controlled trial (STOP-COVID trial, ClinicalTrials.gov number, NCT04469114) on hospitalized patients with COVID-19 pneumonia who were not receiving noninvasive or invasive ventilation found a lower risk of respiratory failure or death through day 28 with tofacitinib [66]. Serious AEs occurred in 20 patients (14.1%) in the tofacitinib group, as compared to 17 (12.0%) in the placebo group, with six cases of transaminases increase (all leading to drug discontinuation), four cases of lymphopenia (all leading to drug discontinuation), and only one case of deep vein thrombosis. Of note, while in the ACTT-2 trial, only approximately 12% of the participants received glucocorticoids; in the STOP-COVID trial the majority (89.3%) of patients were treated with glucocorticoids during hospitalization.

Notwithstanding limitations (289 patients from 15 sites in Brazil, where remdesivir was not available), taken together, these studies open a new avenue for the potential combination of remdesivir, JAK inhibitors, and dexamethasone for treating COVID-19 pneumonia in patients who are not yet receiving invasive mechanical ventilation. In particular, the “window of opportunity” for immune-targeted therapies such as JAK inhibitors might be potentially anticipated to patients with mild to moderate respiratory failure (i.e., before severe organ damage has actually occurred) and with evidence of significant immune activation. Although the optimal timing is uncertain, a recent meta-analysis (six cohort studies and five RCTs) found that JAK inhibitors in people with COVID-19 decreased the use of mechanical ventilation (RR 0.63; 95%CI 0.47–0.84; *p* = 0.002) and increased survival, most convincingly for baricitinib, with borderline impact on rates of ICU admission (0.24; 0.06–1.02; 0.05) and acute respiratory distress syndrome (0.050; 0.19–1.33; 0.16) [67]. To identify these high-risk patients likely to benefit from JAK inhibitors or other immunomodulatory agents, biomarkers such as C-reactive protein, D-Dimer, and IL-6 levels might be used. Results from the multi-arm multi-stage randomized adaptive (AAMMURAVID) trial, supported by AIFA, will hopefully clarify the optimal place in therapy of JAK inhibitors and relevant therapeutic regimen.

From a drug development perspective, direct delivery of JAK inhibition to the lung via inhalation could overcome corticosteroid-resistant pulmonary inflammation while minimizing the potentially excessive systemic immunosuppression. The novel inhaled pan-JAK inhibitor nezulcitinib (TD-0903), designed to target all JAK isoforms, was tested in a phase 2 multiple ascending dose study in 25 hospitalized patients with severe COVID-19: once-daily inhaled administration for 7 days was generally well tolerated, with favorable trends towards improvement in SaO2/FiO2 ratio, respiratory failure-free survival at day 28, and mean time to hospital discharge as compared to placebo (the overall mortality was 33% in placebo-treated patients *versus* 5% in nezulcitinib-treated patients) [68]. A larger (*n* ≈ 200) phase 2 study in hospitalized COVID-19 patients requiring supplemental oxygen is ongoing (NCT04402866).

In the meantime, although no unexpected safety issues emerged from initial real-world experience, including pharmacovigilance data, clinicians should be aware about potential clinically relevant DDIs and the rare occurrence of infections, liver injury, and thrombosis.

## Figures and Tables

**Figure 1 pharmaceuticals-14-00738-f001:**
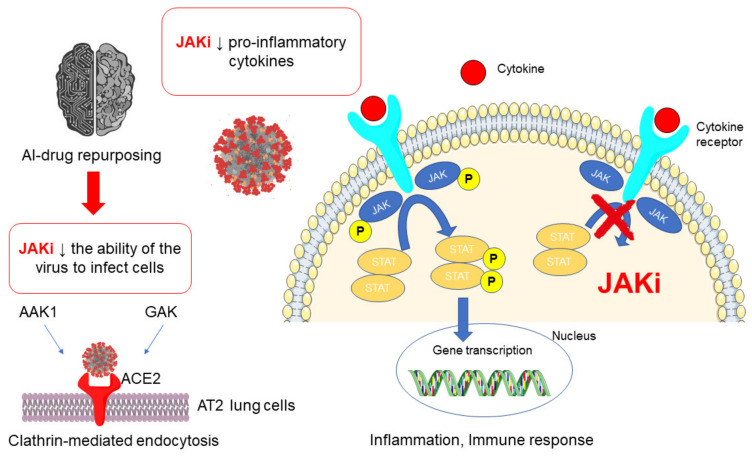
JAK inhibitors (referred as JAKi) in SARS-Cov-2 treatment: rationale and mechanisms of action. AI: artificial Intelligence.

**Table 1 pharmaceuticals-14-00738-t001:** Approved JAK inhibitors.

JAK Inhibitors	Main PK Features	PD Features/JAK Targeting	Approved Indication
**Baricitinib**	Protein binding: 50% t_1/2_: 12 h Metabolism: CYP3A4 Elimination: urinary (69%)/biliary (15%)	ATP competitive kinase inhibitor that selectively, strongly, and reversibly inhibits JAK1 (IC_50_ 5.9 nM) and JAK2 (IC_50_ 5.7 nM)	Rheumatoid arthritis (FDA, EMA)
**Ruxolitinib**	Protein binding: 97% t_1/2_: 3 h Metabolism: CYP3A4 (<50%) and CYP2C9 Elimination: urinary (74%)/biliary (22%)	Selective inhibitor of JAK1 (IC_50_ 3.3 nM) and JAK2 (IC_50_ 2.8 nM)	Myelofibrosis, polycythemia vera (FDA, EMA)
**Fedratinib**	Protein binding: 95% Terminal t_1/2_: 114 h Metabolism: CYP3A4 (major), CYP2C19 and FMOs Elimination: urinary (5%)/biliary (77%)	JAK2-selective inhibitor	Myelofibrosis (FDA, EMA)
**Tofacitinib**	Protein binding: 40% t_1/2_: 3 h Metabolism: CYP3A4 (major), CYP2C19 Elimination: urinary (30%)/biliary (70%)	Potent and selective inhibitor of JAK1 and JAK3	Rheumatoid arthritis, psoriatic arthritis (FDA, EMA), and ulcerative colitis (EMA)
**Upadacitinib**	Protein binding: 52% t_1/2_: 9–14 h Metabolism: CYP3A4 (major), CYP2D6 Elimination: urinary (24%)/biliary (38%)	Selective and reversible inhibitor of JAK1	Rheumatoid arthritis (FDA, EMA)
**Filgotinib**	Protein binding: 55–59% Terminal t_1/2_: 7 h Metabolism: CES2 (major), CES1 Elimination: urinary (87%)/biliary (15%)	Inhibitor of JAK1	Rheumatoid arthritis (EMA)

Abbreviations: ATP: adenosine triphosphate; CES: carboxylesterases; CYP: cytochrome P450; EMA: European Medicines Agency; FDA: Food and Drug Administration; IC_50_: 50% inhibitory concentrations; PD: pharmacodynamics; PK: pharmacokinetics; FMOs: flavin-containing monooxygenases; and t_1/2_: half-life.

**Table 2 pharmaceuticals-14-00738-t002:** Summary of the clinical studies investigating efficacy and safety of baricitinib in COVID-19 patients.

Study Reference	Study Design	No. Patients	Intervention Group	Comparator Group	Primary Outcome	Secondary Outcome	Safety Assessment
Kalil et al., 2021 [28]	RCT, double-blind, multicenter (USA, Singapore, South Korea, Mexico, Japan, Spain, Denmark, and UK)	1033	Baricitinib + Remdesivir (*n* = 515) Baricitinib: 4 mg/day for 14 days Remdesivir: 200 mg day 1–100 mg day 2–10	Placebo + Remdesivir (*n* = 518) Remdesivir: 200 mg day 1–100 mg day 2–10	Median time to recovery: 7 days versus 8 days (RR 1.16; CI 1.01–1.32; *p* = 0.03) Median time to recovery (patients receiving non-invasive ventilation or high-flow oxygen): 10 days versus 18 days (RR 1.51; CI 1.10–2.08) No difference in median time to recovery in patients receiving mechanical ventilation or ECMO	Clinical status at day 15 (odds of improvement): OR 1.3 (CI 1.0–1.6) 14-day mortality rate: 1.6% versus 3.0% (HR 0.54; CI 0.23–1.28) 28-day mortality rate: 5.1% versus 7.8% (HR 0.65; CI 0.39–1.09) Incidence of new use of oxygen: (−17.4%; CI −31.6 to −2.1) Incidence of new use of invasive ventilation or ECMO: (−5.2%; CI −9.5 to −0.9)	Serious AEs 16.0% versus 21.0% (difference −5.0 percentage points; CI −9.8 to −0.3; *p* = 0.03) New infections 5.9% versus 11.2% (difference −5.3 percentage points; CI −8.7 to −1.9; *p* = 0.003)
Stebbing et al., 2021 [25]	Observational, prospective, multicenter, propensity score matching	166	Baricitinib 2–4 mg/day (*n* = 83)	Standard of care (Hydroxychloroquine + Lopinavir/Ritonavir + Corticosteroids) (*n* = 83)	Primary composite endpoint of death or invasive mechanical ventilation: 16.9% versus 34.9% (*p* < 0.001)	Baricitinib was independently associated as a protective variable with the primary outcome at multivariate regression analysis (HR 0.29; CI 0.15–0.58; *p* < 0.001)	Transaminase increase (19%) Bacterial infection(14%)
Hasan et al., 2020 [27]	Observational, prospective, case–control (Bangladesh)	37 COVID-19 moderate-severe pneumonia	Baricitinib 4 mg/day after 8 mg LD (*n* = 20)	Baricitinib 4 mg/day without LD (*n* = 17)	Median days required to stop the need of supplement oxygen: 5 versus 8 (*p* = 0.001) ICU admission rate: 10% versus 29.4% (*p* = 0.005)	Median length of hospital stay: 12 versus 15 (*p* = 0.028) No difference in 30-day mortality rate	-
Rodriguez-Garcia et al., 2020 [29]	Observational prospective (Spain)	112 COVID-19 moderate-severe pneumonia	Baricitinib + standard of care (Hydroxychloroquine + Lopinavir/Ritonavir + corticosteroids) (*n* = 62)	Standard of care (*n* = 50)	Improvement in SpO2/FiO2: mean difference 49 (*p* < 0.001) No difference in mortality and ICU admission rate	Proportion of patients required supplemental oxygen: risk reduction of 82% and 69%, respectively at discharge (*p* < 0.001) and at 1 month (*p* = 0.024)	-
Cantini et al., 2020 [22]	Observational retrospective, multicenter, longitudinal (Italy)	191 COVID-19 moderate pneumonia	Baricitinib 4 mg/day for 14 days + Lopinavir/Ritonavir 250 mg × 2/day for 14 days (*n* = 113)	Hydroxychloroquine 400 mg/day + Lopinavir/Ritonavir 250 mg × 2/day (*n* = 78)	14-day mortality rate: 0.0% versus 6.4% (*p* = 0.01)	ICU admission: 0.88% versus 17.9% (*p* = 0.019) Hospital discharge rate (at 2 weeks): 77.8% versus 12.8% (*p* < 0.0001)	Transaminase increase (3.5%)
Cantini et al., 2020 [17]	Observational retrospective (Italy)	24 COVID-19 moderate pneumonia	Baricitinib 4 mg/day for 14 days + Lopinavir/Ritonavir 250 mg × 2/day for 14 days (*n* = 12)	Hydroxychloroquine 400 mg/day + Lopinavir/Ritonavir 250 mg × 2/day (*n*= 12)	Hospital discharge at 2-week:58% versus 8% (*p* = 0.027)	Significant improvement in P/F ratio and CRP levels	No serious AEs One case of baricitinib withdrawal
Bronte et al., 2020 [21]	Observational retrospective, Longitudinal (Italy)	76	Baricitinib 4 mg × 2/day days 1–2 + 4 mg/day days 3–9 + standard of care (*n* = 20)	Standard of care (Hydroxychloroquine + Lopinavir/Ritonavir) (*n* = 56)	Mortality rate: 5% versus 45% (*p* < 0.001) No difference in ARDS incidence or disease duration	Faster reduction in the need for oxygen flow therapy (*p* < 0.001) and a more rapid increase in the P/F ratio compared with the control group (*p* = 0.02), as well as a reduction in serum levels of CRP (*p* < 0.001)	-
Rosas et al., 2020 [24]	Observational retrospective (Spain)	60	Baricitinib 2–4 mg/day ± Tocilizumab 400–600 mg single dose (*n* = 23)	Standard of care ± Tocilizumab (*n* = 37)	No difference in mortality or ICU admission rate	Significant reduction in mean respiratory rate at discharge (20 versus 24; *p* < 0.05)	No serious AEs
Titanji et al., 2020 [26]	Observational retrospective cohort, non-controlled (USA)	15 COVID-19 moderate-severe pneumonia	Baricitinib 2–4 mg/day + Hydroxychloroquine 200–400 mg/day	-	ICU admission: 60% Overall mortality rate: 20% Reduction in CRP level: 86.7% Recovery rate: 80%Clinical improvement: 73.3%	-	-
Stebbing et al., 2020 [14]	Case series (Italy)	4	Baricitinib 2–4 mg/day for 10–12 days	-	Moderate-severe disease: 75% Clinical improvement: 100%	-	Transient increase in serum transaminases in all four patients
Cingolani et al., 2020 [23]	Case report (Italy)	1	Baricitinib 4 mg × 2/day for 14 days after sub-intensive care unit admission	Failure to standard of care (Lopinavir/Ritonavir + Hydroxychloroquine + Azithromycin) + Sarilumab 400 mg on day 1 and 4	Constant increase in the pO2 coupled with progressive decrease in required FiO2	-	-

AE: adverse event; ARDS: acute respiratory distress syndrome; CI: confidence interval; CRP: C-reactive protein; ECMO: extracorporeal membrane oxygenation; HR: hazard ratio; ICU: intensive care unit; LD: loading dose; OR: odds ratio; and RR: relative risk.

**Table 3 pharmaceuticals-14-00738-t003:** Summary of the clinical studies investigating efficacy and safety of ruxolitinib in COVID-19 patients.

Study Reference	Study Design	No. Patients	Intervention Group	Comparator Group	Primary Outcome	Secondary Outcome	Safety Assessment
Cao et al., 2020 [31]	Randomized controlled, multicentre, single-blind (China)	43 severe COVID-19 pneumonia	Ruxolitinib 5 mg × 2/day + standard of care (antivirals + corticosteroids and supportive treatment) (*n* = 22)	Placebo + standard of care (antivirals + corticosteroids and supportive treatment) (*n* = 21)	Median time to clinical improvement: 12 days versus 15 days (*p* = 0.147)	Significant improvement on chest CT scan at 14 days: 90% vs. 61.9% (*p* = 0.0495) 28-day mortality rate: 0.0% vs. 14.3% (*p* = 0.232)	No difference in serious AEs
Giudice et al., 2020 [36]	Observational prospective, monocentric (Italy)	17 severe COVID-19 pneumonia	Ruxolitinib 10 mg × 2/day for 14 days + Eculizumab 900 mg/week (*n* = 7)	Best available therapy (Hydroxychloroquine + Azithromycin + Heparin) (*n* = 10)	Improvement in median PaO2 after 7 days: 94 versus 77 (*p* = 0.026) Improvement in median PaO2/FiO2 after 7 days: 370.5 versus 246 (*p* = 0.0395)	No significant difference in mortality rate and duration of hospitalization	Increase in transaminase levels: 71.4%
Vannucchi et al., 2020 [41]	Observational prospective, monocentric (Italy)	34	Ruxolitinib 5–10 mg × 2/day	-	Clinical improvement in 85.3% of cases (reduction of at least 2 points in seven-point ordinal scale) Less frequent clinical improvement in patients with more severe respiratory impairment HR 0.31 (CI 0.1–1.0)	CRP levels significantly decreased from a baseline median level of 72 mg/l (IQR, 39–111) to 26 mg/l (IQR, 5–76; *p* = 0.03) by day 7 and normalized by day 14 (12 mg/l, IQR, 6–21; *p* < 0.001)	Discontinuation of treatment in 14.7%
Mortara et al., 2021 [42]	Observational prospective, monocentric (Italy)	31	Ruxolitinib 5 mg × 2/day for 15 days	-	Improvement in symptoms (Likert scale) at 7 and 15 days: 80.6% and 90.3%	-	No AEs observed during treatment
Capochiani et al., 2020 [32]	Observational retrospective cohort, multicenter (Italy)	18 ARDS due to COVID-19	Ruxolitinib 20 mg × 2/day in day 1–2, 5–10 mg × 2/day up to day 14 (*n* = 18)	-	No evolution from NIV to mechanical ventilation: 88.9% Significant improvement in respiratory response within 48 h: 88.9%	14-day complete respiratory function: 88.9% Rapid restoration within 48 h in PaO2/FiO2: 88.9%	No AEs observed during treatment and at the follow-up
La Rosée et al., 2020 [39]	Observational retrospective, monocentric (Germany)	14 severe COVID-19 pneumonia	Ruxolitinib 7.5 mg × 2/day with subsequent reassessment for increase or decrease in dosage	-	Reduction by 25% in COVID-19 inflammatory score achieved after 5 days	-	One patient transient grade 3 liver toxicityTwo patients experienced grade 3 anaemia
Gaspari et al., 2020 [35]	Case report (Italy)	2	(a) Ruxolitinib 5 mg × 2/day on day 1–2 and 10 mg × 2/day on day 3–5 (b) Ruxolitinib 5 mg × 2/day on day 1–7	-	-	-	(a) Skin purpuric lesion associated with reduction in platelet count (b) Erythrodermic rash on whole body surface
Sammartano et al., 2020 [40]	Case report (Italy)	1	Ruxolitinib 20 mg × 2/day	Prior clinical failure with therapy including Hydroxychloroquine, Azithromycin, Corticosteroids, and Tocilizumab	COVID-19 related ARDS in a patient with diagnosis of Blastic Plasmocitoid Dendritic Cell Neoplasm Clinical improvement after 48 h with CPAP discontinuation	-	-
Saraceni et al., 2021 [43]	Case report (Italy)	1	Ruxolitinib 5 mg × 2/day for chronic GVHD after allogeneic stem cell transplant, discontinued at hospital admission for COVID-19 pneumonia and re-started after clinical failure with standard of care	Hydroxychloroquine, Lopinavir-Ritonavir, and LMWH	Rapid improvement in respiratory function and hospital discharge at day 45	-	No reported AEs
Innes et al., 2020 [37]	Case report (UK)	1	Ruxolitinib 5 mg × 2/day on day 1 × 3 and 10 mg × 2/day on day 4 × 21	Prior clinical failure with intermediate dosage of LMWH and tocilizumab	Improvement in respiratory function and hospital discharge at day 28	-	No reported AEs
Koschmieder et al., 2020 [38]	Case report (Germany)	1	Ruxolitinib 10 mg × 2/day chronic treatment (since 15 months) for myelofibrosis	-	ICU admission (no required mechanical ventilation) Rapid improvement in respiratory function and hospital discharge at day 15	-	No reported AEs
Caradec et al., 2020 [33]	Case report (France)	1	Ruxolitinib 10 mg × 2/day	Prior clinical failure with Hydroxychloroquine + Azithromycin	Improvement in respiratory function after 48 h and CRP normalization at day 8	-	No reported AEs
Foss et al., 2020 [34]	Case report (USA)	1	Ruxolitinib 10 mg × 2/day for chronic GVHD after allogeneic stem cell transplant	-	Attenuated COVID-19 infection in an immunosuppressed patient in chronic treatment with ruxolitinib	-	-
Betelli et al., 2020 [30]	Case report (Italy)	1	Ruxolitinib 5 mg × 2/day for 14 days + Dexamethasone 20 mg/day for 5 day and subsequent decalage	Prior clinical failure with standard of care (Hydroxychloroquine + Azithromycin)	Oxygen supplementation suspended after 14 days Hospital discharge after 23 days	-	-

AE: adverse event; ARDS: acute respiratory distress syndrome; CI: confidence interval; CPAP: continuous positive airway pressure; CRP: C-reactive protein; CT: computed tomography; GVHD: graft-versus-host disease; HR: hazard ratio; IQR: interquartile range; LMWH: low-molecular weight heparin; and NIV: non-invasive ventilation.

**Table 4 pharmaceuticals-14-00738-t004:** Predicted pharmacokinetic drug interactions between JAK inhibitors and other agents used for the management of COVID-19.

Pharmacokinetic Feature		DDIs with JAK-Inhibitors		Clinical Relevance and Literature Data
		Baricitinib	Ruxolitinib	
**P-gp substrate**		-	Weak inhibitor	
**CYP3A4 substrate**		Minor (only 10%)	Major (CYP2C9/CYP2D9 minor)	
**BCRP substrate**		Substrate Weak inhibitor (only in vitro)	Weak inhibitor	
**OAT substrate**		OAT3 substrate OAT1/3 inhibitor (only in vitro)	Weak inhibitor (only in vitro)	
**COVID-19 agents**	**Metabolic pathway**			
**Remdesivir**	CYP2C8–CYP2C19–CYP3A4–P-gp–OATP1B1 substrate			No relevant interactions expected
**Dexamethasone**	CYP3A4 substrate–moderate CYP3A4 inducer			No relevant interactions expected
**Colchicine**	CYP3A4 and P-gp substrate			Risk of increased colchicine exposure with concomitant use of ruxolitinib, particularly in patients with renal or hepatic impairment
**IL6 inhibitors**	Restoration of CYP3A4 and CYP2C19 activity			Risk of additive immunosuppression
**Favipiravir**	CYP2C8, OAT1, and OAT3 moderate inhibitor			Favipiravir may increase baricitinib exposure, but not in a clinically relevant extent

RED BOX: avoid co-administration (contraindicated or not recommended). ORANGE BOX: potential interaction (caution should be exercised and consider dose adjustment or alternative drugs). YELLOW BOX: potential weak interaction (monitoring for potential underexposure or toxicity). GREEN BOX: no interaction expected based on pharmacokinetic properties, although no clinical data exist. DDIs were checked through covid19-druginteractions.org/checker. BCRP: breast cancer resistance protein; CYP: cytochrome P450; OAT: organic anion transporter; and P-gp: glycoprotein P.

**Table 5 pharmaceuticals-14-00738-t005:** Cardiovascular profile of t drugs used for the management of COVID-19. Only drugs with “most beneficial” effect on mortality or mechanical ventilation, duration of ventilation, length of hospital stay, or time to resolution of symptoms (as compared to the standard of care) were selected based on the updated living systematic review [1], last access 4 June 2021.

Drug	QT prolongation */arrhythmia	Myo-pericarditis	MACE	VTE
**JAK inhibitors**				
**Remdesivir**	§			
**Dexamethasone**				
**Colchicine**	#			
**IL6 inhibitors**				
**Favipiravir**				

* crediblemeds.org (accessed 4 June 2021); § bradycardia; # potential reduction in atrial fibrillation; VTE: venous thromboembolism; MACE: major adverse cardiovascular events. Please note that the definition of this composite endpoint may vary among studies, but usually includes cardiovascular death, myocardial infarction, ischemic stroke; GREY BOX: unknown effect (no dedicated study performed); WHITE BOX: no effect (in a dedicated study); YELLOW BOX: unclear effect; RED BOX: negative effect (increased risk); and GREEN BOX: positive effect (reduced risk).

**Table 6 pharmaceuticals-14-00738-t006:** Case-by-case assessment of reports concerning adverse events of clinical interest with baricitinib for COVID-19 management.

Toxicity Areas	AE of Clinical Interest Reported with Baricitinib	No. Patients	No. Deaths	Proportion of Death	Seriousness	Mean Age	Sex	Weight (Kg)	Concomitant Drugs Used for COVID-19 Management	Other Relevant Concomitant Medications Implicated in Specific AE of Clinical Interest
**Cardiotoxicity**	Hypotension (6) Bradycardia (2) Tachycardia (1) Ventricular extrasystoles (1) Atrial fibrillation (1) Ventricular tachycardia (1)	8	6	75.0%	100.0%	61.9 ± 16.6	4 M–4 F	94.7 ± 28.1	8 Remdesivir 8 Dexamethasone 1 COVID-19 convalescent plasma1 Methylprednisone	-
**Hepatotoxicity**	Alanine aminotransferase increased (5) Aspartate aminotransferase increased (5) Liver function test increased (3) Drug-induced liver injury (1) Liver disorder (1) Transaminase increased (1)	9	0	0.0%	100.0%	64.4 ± 12.8	4 M–5 F	97.0 ± 37.4	8 Remdesivir 6 Dexamethasone 1 Enoxaparin	1 Azithromycin 1 Ceftriaxone 1 Cefepime 1 Fluconazole
**Infection**	Septic shock (7) Sepsis (2) Infection (3) Pneumonia staphylococcal (4) Urinary tract infection (2) Candida infection (2) Pneumonia (2) Staphylococcal infection (2) Bacterial infection (2) Candida test positive (2) Other infections *	25	8	32.0%	100.0%	59.4 ± 16.0	13 M–12 F	102.5 ± 35.6	25 Remdesivir 21 Dexamethasone 5 Methylprednisone 2 Heparin 2 COVID-19 convalescent plasma1 Prednisone	-
**Nephrotoxicity**	Acute kidney injury (11) Renal impairment (3) Renal failure (1) Renal disorder (1)	16	9	56.3%	100.0%	62.1 ± 14.6	7 M–9 F	94.9 ± 23.7	15 Remdesivir 14 Dexamethasone 2 Methylprednisone 1 Prednisone 1 Heparin	3 Piperacillin-tazobactam 1 Vancomycin
**Thromboembolic events**	Deep vein thrombosis (4) Pulmonary embolism (4) Embolism (2) Vena Cava thrombosis (1) Thrombosis (1) Embolism venous (1)	13	5	38.5%	100.0%	65.2 ± 16.4	6 M–6 F–1 NS	102.2 ± 50.0	10 Remdesivir 6 Dexamethasone 2 Methylprednisone 2 Enoxaparin 2 Heparin	-

AE: adverse event; NS: not specified; * Other infections: Pneumonia pseudomonal, Pneumonia serratia, Pneumococcal infection, Escherichia urinary tract infection, Corynebacterium infection, Pseudomonas infection, Urinary tract infection pseudomonal, Urinary tract infection enterococcal, Klebsiella infection, Streptococcal infection, Stenotrophomonas test positive, Streptococcus test positive, Klebsiella test positive, Bacteraemia, and Pneumonia bacterial (one each).

## Data Availability

Data sharing not applicable.

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
