# Peer review of "Janus Kinase Inhibitors and Coronavirus Disease (COVID)-19: Rationale, Clinical Evidence and Safety Issues"

_pharmaceuticals, 2021, doi:10.3390/ph14080738_

Round 1
Reviewer 1 Report
The article summarizes recent research about the role of JAK inhibtors in COVID-19. The JAK inhibitors are kind of novel drugs that represent promising medicines targeting JAK to treat inflammatory and autoimmune disease. Cytokine storm is one of a key factor in the aggravation of the COVID-19, and most cytokines act through JAK-STAT signal pathway.
The paper is timely and will be of interest to broad audience of students and established scientists alike. I think that authors made a good job putting together the most relevant literature.
Although JAK inhibitors for COVID-19 make sense in theory, there have been too few clinical trials so far. It would enrich our conclusions if we could add some data on COVID-19 treatment with unmarketed JAK inhibitors, such as clinical Phase 2 trail of Nezulcitinib (NCT04402866, Theravance Biopharma).
Author Response
Reviewer#1
The article summarizes recent research about the role of JAK inhibtors in COVID-19. The JAK inhibitors are kind of novel drugs that represent promising medicines targeting JAK to treat inflammatory and autoimmune disease. Cytokine storm is one of a key factor in the aggravation of the COVID-19, and most cytokines act through JAK-STAT signal pathway.
The paper is timely and will be of interest to broad audience of students and established scientists alike. I think that authors made a good job putting together the most relevant literature.
Although JAK inhibitors for COVID-19 make sense in theory, there have been too few clinical trials so far. It would enrich our conclusions if we could add some data on COVID-19 treatment with unmarketed JAK inhibitors, such as clinical Phase 2 trial of Nezulcitinib (NCT04402866, Theravance Biopharma).
Thank you for the appreciation. We agree with the reviewer on the importance of providing a broad perspective on drug development, which is now addressed in the conclusion, also by adding relevant Phase 2 study on nezulcitinib (Eur Respir J. 2021 Jul 1:2100673. doi: 10.1183/13993003.00673-2021).
Reviewer 2 Report
This paper is a comprehensive review on the role of the safety and efficacy of JAK inhibitors as repurposed drugs in the treatment of severe COVID19. PK and PD data are well articulated, across several molecules, and the evidence have been retrieved from the literature based on a research string for systematic review. The evidence synthesis is clear and well presented, in tables. The topic is relevant and timely, as the decision of EMA in Europs for the use of baricitinib is expected to be disclosed in weeks, here.
A big review has focused on baricitinib and has been published on one nature journal, last May: https://www.nature.com/articles/s41375-021-01266-6. However, the scope and contents of this Chinese paper is more limited than the one submitted to your Journal, and only focusing on a single molecule. Therefore, this paper adds more to the body of the literature, from a very comprehensive pharmacological perspective.
Paper is very linear, the discussion flows very clearly.
The conclusions, the evidence and arguments presented are total coherence, the authors well supported their statement in an evidence-based manner.
Author Response
Reviewer#2
This paper is a comprehensive review on the role of the safety and efficacy of JAK inhibitors as repurposed drugs in the treatment of severe COVID19. PK and PD data are well articulated, across several molecules, and the evidence have been retrieved from the literature based on a research string for systematic review. The evidence synthesis is clear and well presented, in tables. The topic is relevant and timely, as the decision of EMA in Europs for the use of baricitinib is expected to be disclosed in weeks, here.
A big review has focused on baricitinib and has been published on one nature journal, last May: https://www.nature.com/articles/s41375-021-01266-6. However, the scope and contents of this Chinese paper is more limited than the one submitted to your Journal, and only focusing on a single molecule. Therefore, this paper adds more to the body of the literature, from a very comprehensive pharmacological perspective.
Paper is very linear, the discussion flows very clearly.
The conclusions, the evidence and arguments presented are total coherence, the authors well supported their statement in an evidence-based manner.
Thank you for the appreciation. The conclusion was implemented by adding the relevant meta-analysis quoted by the Reviewer, and also by providing a broad perspective on drug development (Phase 2 study on nezulcitinib).